# Toward Self-Evolving Systems of LLM Agents through Exploration and Iterative Feedback

## Abstract

Training large language model (LLM) agents to acquire necessary skills and perform diverse tasks within an environment is gaining interest as a means to enable open-endedness. However, creating the training dataset for their skill acquisition faces several challenges. Manual trajectory collection requires significant human effort. Another approach, where LLMs directly propose tasks to learn, is often invalid, as the LLMs lack knowledge of which tasks are actually feasible. Moreover, the generated data may not provide a meaningful learning signal, as agents often already perform well on the proposed tasks. To address this, we propose a novel framework, **EX**ploration and **I**terative **F**eedback (**EXIF**), for LLM-powered agents. This automatic improvement framework is designed to enhance the feasibility of generated target behaviors while accounting for the agents' capabilities. Our method adopts an exploration-first strategy by employing an exploration agent (`Alice`) to train the target agent (`Bob`) to learn essential skills in the environment. Specifically, `Alice` first interacts with the environment to generate a feasible, environment-grounded skill dataset, which is then used to train `Bob`. Crucially, we incorporate an iterative feedback loop, where `Alice` evaluates `Bob`'s performance to identify areas for improvement. This feedback then guides `Alice`'s next round of exploration, forming a closed-loop data generation process. Experiments on Webshop and Crafter demonstrate **EXIF**'s ability to iteratively expand the capabilities of the trained agent without human intervention, leading to substantial performance improvements. Interestingly, we observe that setting `Alice` to the same model as `Bob` also notably improves performance, demonstrating **EXIF**'s potential for building a self-evolving system.

## 1 Introduction

Large language model (LLM)-powered agents have demonstrated remarkable capabilities in interacting with complex environments and performing user-instructed tasks, including game playing (Wang et al., 2023; Hu et al., 2024) and graphical user interface (GUI) manipulation (Zhou et al., 2024a; Xie et al., 2024; Lee et al., 2024; Rawles et al., 2025). A significant aspiration for these agents is to achieve open-endedness: the ability to autonomously explore, learn, and continuously expand their capabilities within an environment, effectively becoming capable of tackling an ever-growing range of tasks without human intervention. This kind of open-endedness cannot be easily achieved with prompting techniques such as reasoning (Yao et al., 2023), reflection (Shinn et al., 2023), and tree search (Koh et al., 2024). These in-context learning mechanisms are often insufficient for fostering continuous, autonomous learning—especially in unfamiliar settings where the agent lacks awareness of possible actions and their consequences (Chen et al., 2023; Zeng et al., 2023; Zhou et al., 2024c), necessitating continuous learning mechanisms within the environment.

To cultivate open-ended learning and enable agents to continuously acquire specialized skills in new environments, collecting suitable training data is a critical step. A straightforward approach is to manually collect instructions and corresponding trajectories for a multitude of potential tasks in each environment, but this is often infeasible due to high costs. Consequently, recent work harnesses the generative capabilities of LLMs to automatically synthesize instruction-trajectory datasets (Murty et al., 2024b; Pahuja et al., 2025), reducing human annotation effort and enabling scalable data collection across diverse environments. These methods often prompt LLMs to directly propose tasks

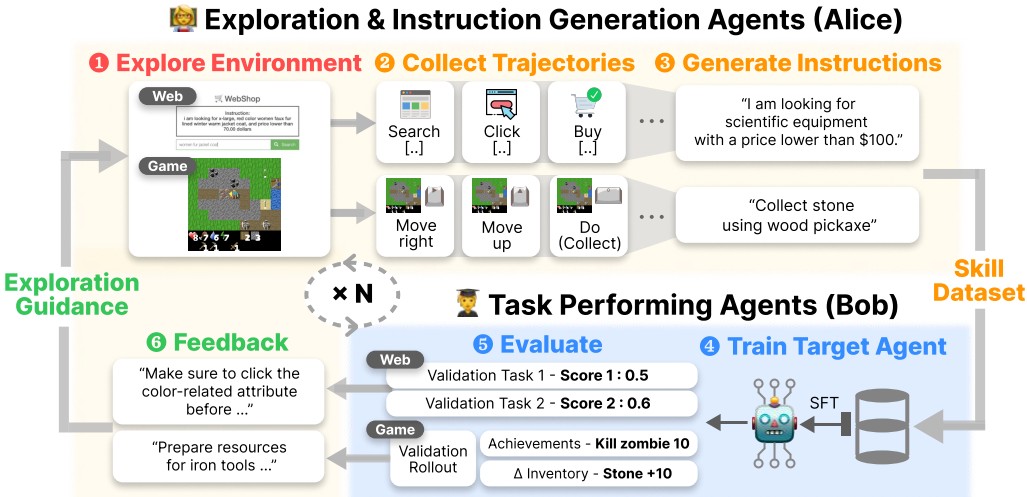

Figure 1: Overview of our framework for self-evolving systems through exploration and iterative feedback (**EXIF**), consisting of two main components: **(1)** an **explore-first strategy** that enables the agent, **Alice**, to navigate the environment and generate feasible, valid tasks, which are then used to train another agent, **Bob**; and **(2)** an **iterative feedback** mechanism that produces tasks and trajectories beyond **Bob**'s current capabilities to expand its skills. Through multiple iterations, **EXIF** enables **Bob** to expand its skill set in the target environment without any human guidance.

and then collect trajectories conditioned on those tasks—a process we refer to as *proposal-first* task generation (Zhou et al., 2024b; Zheng et al., 2025; Su et al., 2025; Zhao et al., 2025).

However, applying this proposal-first approach to foster open-ended learning presents two critical downsides. *First*, without actively interacting with the environment, LLMs cannot determine which tasks are feasible when making their proposals, potentially generating a large volume of invalid tasks. *Second*, lacking awareness of the current agent's evolving capabilities during its training lifecycle, LLMs may produce synthetic data that is misaligned with what the agent actually needs to learn to expand its skill set effectively. Because these requirements are unmet, much of the resulting synthetic data may be irrelevant or suboptimal, failing to effectively guide the agent toward learning the essential skills in the target environment (Murty et al., 2024b; He et al., 2024a; Yuan et al., 2023).

In this paper, we propose a novel self-evolving system for language agents, based on **EX**ploration and **I**terative **F**eedback (**EXIF**). Our method integrates two crucial components: **(a)** exploration-based skill dataset generation and **(b)** multi-iteration feedback. **EXIF** utilizes two LLM agents: **Alice**, which generates exploratory trajectories and corresponding instructions—pairing them into a *skill dataset*, referring to data used to learn necessary skills in the environment—and **Bob**, which is trained on this dataset to effectively perform tasks in the given environment. Specifically, **Alice** explores the environment and converts these explorations into feasible trajectories and instructions. This ensures that the generated tasks are grounded in the environment, unlike proposal-first approaches, which risk producing infeasible tasks. **Bob** is then trained on the generated dataset. Subsequently, **EXIF** incorporates an iterative feedback loop: **Alice** identifies areas where **Bob** struggles and provides targeted feedback. Based on this feedback, **Alice** generates a new, tailored skill dataset to address these specific needs. As a result, **EXIF** iteratively enhances **Bob**'s skill repository by grounding skills in both the environment and **Bob**'s own capabilities, enabling continual evolution and generalization to unseen tasks without human intervention.

Through extensive experiments on two challenging benchmarks, Webshop (Yao et al., 2022) and Crafter (Hafner, 2022), we show that **EXIF** results in a consistent improvement of LLM agent over the iterative training process. Specifically, in Webshop, when **Alice** is GPT-4o (Hurst et al., 2024), the LLM agent trained with **EXIF** improves its reward substantially, from 2.0 to 52.6 over training iterations; in Crafter, it achieves performance comparable to GPT-4o. Moreover, we demonstrate that even when using the same small model for both **Alice** and **Bob**, the approach yields notable performance improvements, including a 15% higher success rate in Webshop compared to when

**Alice** is the larger GPT-4o model, highlighting the potential for building self-evolving systems. We believe that our method paves a way for more autonomous, self-improving AI agents that learn and adapt in complex environments with minimal human guidance, enabling a new generation of intelligent systems.

## 2 METHOD

In this section, we introduce **EXIF**, a novel algorithm for self-evolving systems through exploration and iterative feedback. As illustrated in Figure 1, **EXIF** employs an LLM agent, hereafter referred to as **Alice** (with policy $\pi_\phi$), for exploration-based skill dataset generation and feedback processing. These skill datasets from **Alice** are then used to iteratively train a target LLM agent, hereafter referred to as **Bob** (with policy $\pi_\theta$). Therefore, **Alice** iteratively creates a **skill dataset** conditioned on natural language feedback about **Bob**'s performance, and **Bob** is **iteratively refined** to perform tasks well in the given environment. The pseudocode is in Appendix C, the prompts for **Alice** and **Bob** in Appendix D, and additional implementation details in Appendix E.

Throughout, we consider an agent interacting with an environment over discrete time steps $t = 1, 2, \ldots, T$, receiving observation $o_t \in \mathcal{O}$ and taking action $a_t \in \mathcal{A}$ based on the history $h_t = (o_{t-H}, a_{t-H}, \ldots, o_{t-1})$ and optionally a goal $g$. We use an LLM as a policy $\pi_\phi$ (or $\pi_\theta$), producing actions as $a_t \sim \pi_\phi(\cdot \mid h_t, o_t, g)$. The full trajectory is denoted $\tau = (o_1, a_1, \ldots, o_T, a_T)$.

Specifically, our method consists of the following steps:

- **Step 1 (Exploration & Skill Dataset Generation)**: **Alice** explores the target environment to collect diverse trajectories, then generates instructions from them to create synthetic instruction–trajectory pairs (*skill dataset*) (Section 2.1).
- **Step 2 (Training Target Agent & Evaluation)**: The generated skill dataset is used to fine-tune **Bob**, which is then evaluated in the target environment (Section 2.2).
- **Step 3 (Feedback & Repeat (Steps 1–3))**: **Alice** provides feedback on **Bob**'s evaluation and repeats Steps 1–3, with exploration (Step 1) now conditioned on this feedback to enable targeted data generation for **Bob** in subsequent rounds of fine-tuning (Section 2.3).

### 2.1 EXPLORATION

The initial phase focuses on gathering diverse behavioral data from the environment using **Alice**'s policy $\pi_\phi$. Unlike typical goal-oriented agents, **Alice** operates without an explicit external goal $g$ during this phase. This is because **Alice** often lacks prior knowledge of the environment, and exploring with an arbitrary goal, proposed by **Alice**, might lead to invalid trajectories if the goal is not achievable within the environment.

Specifically, **Alice** interacts with the environment over time steps $t = 1, \ldots, T$, generating actions $a_t \sim \pi_\phi(\cdot|h_t, o_t)$ based solely on the interaction history $h_t = (o_{t-H}, a_{t-H}, \ldots, o_{t-1})$ and the current observation $o_t$. The objective is to produce a wide range of interaction sequences or trajectories, $\tau_{exp} = (o_1, a_1, \ldots, o_T, a_T)$, capturing various feasible behaviors within the environment's constraints. To avoid excessive random behavior, we use weak constraints such as assigning a persona during exploration or setting a vague objective like survival in the game environment. Exploration continues until a termination condition is met (e.g., reaching a maximum step count $T_{max}$). This process yields an initial dataset of exploratory trajectories $\mathcal{D}_{exp} = \{\tau_{exp}^{(j)}\}_{j=1}^M$.

**Exploration with Feedback**    *After the first iteration*, exploration is conditioned on feedback from the previous iteration $k$ (detailed in Section 2.3). The feedback $F^{(k)}$ guides **Alice** in generating a new skill dataset for the next round, $k + 1$, specifically tailored to address the shortcomings identified in **Bob** during iteration $k$. **Alice**'s action is now conditioned on the feedback: $a_t \sim \pi_\phi(\cdot \mid h_t, o_t, F^{(k)})$, steering exploration toward behaviors and states relevant to the skills **Bob** lacks.

**Instruction Generation**    To train **Bob**, we convert exploratory trajectories from **Alice** into a skill dataset. **Alice** analyzes each trajectory $\tau_{exp}^{(j)}$ and generates a natural language instruction $I^{(j)}$ that describes the demonstrated task or behavior. This yields the final skill dataset $\mathcal{D}_{skill} = \{(I^{(j)}, \tau^{(j)})\}_{j=1}^M$, where each instruction $I^{(j)}$ is grounded in a corresponding trajectory $\tau_{exp}^{(j)}$.

## 2.2 FINE-TUNING BOB

The generated dataset $\mathcal{D}_{skill}$ is used to train the target agent, **Bob**, whose policy $\pi_\theta$ is parameterized by $\theta$. We employ supervised fine-tuning (SFT) to teach **Bob** ($\pi_\theta$) to execute the generated instructions $I^{(j)}$ by mimicking the actions $a_t^{(j)}$ in the corresponding trajectories $\tau^{(j)} = (o_1^{(j)}, a_1^{(j)}, \ldots, o_{T_j}^{(j)}, a_{T_j}^{(j)})$. Specifically, **Bob** ($\pi_\theta$) is trained to maximize the likelihood of the actions in the trajectory given the instruction and the history. This is achieved by minimizing the SFT loss over the dataset $\mathcal{D}_{skill}$:

$$\mathcal{L}_{SFT}(\theta; \mathcal{D}_{skill}) = -\sum_{j=1}^{M} \sum_{t=1}^{T_j} \log \pi_\theta(a_t^{(j)} | h_t^{(j)}, o_t^{(j)}, I^{(j)}), \tag{1}$$

where $h_t^{(j)} = (o_{t-H}^{(j)}, a_{t-H}^{(j)}, \ldots, o_{t-1}^{(j)})$ is the history at $t$ with context length $H$ within trajectory $j$. This initial training yields the first version of **Bob**'s fine-tuned policy $\pi_{\theta^{(0)}}$.

## 2.3 FEEDBACK GENERATION & ITERATIVE PROCESS

**EXIF** incorporates an iterative refinement loop (indexed by $k = 0, 1, 2, \ldots$) to progressively enhance **Bob**'s ($\pi_\theta$) capabilities by targeting areas for improvement. Each iteration involves evaluating **Bob** at iteration $k$, generating targeted data using **Alice** ($\pi_\phi$) guided by feedback for the next iteration $(k + 1)$, and retraining **Bob** ($\pi_{\theta^{(k)}}$).

**Feedback Generation**    To generate feedback for iteration $k+1$, the performance or behaviors of the current **Bob** policy $\pi_{\theta^{(k)}}$ in the target environment are evaluated. This evaluation involves executing **Bob** on a set of evaluation tasks or allowing it to interact within the environment, potentially attempting tasks similar to those in the training set or novel ones. Analyzing its successes and failures—such as the inability to follow certain instructions or failure to complete specific sub-tasks as reflected in the $o_t, a_t$ sequences—then yields a natural language feedback signal $F^{(k)}$. This signal encodes the deficiencies or areas where **Bob** ($\pi_{\theta^{(k)}}$) requires improvement.

**Repeat the Process**    After feedback generation, the next iteration begins: exploration and instruction generation with **Alice**, fine-tuning **Bob**, evaluation, and feedback generation. Note that **Alice**'s parameter is not updated during this process. The only key difference starting from iteration 1 is that the first step—exploration—is now conditioned on the feedback signal $F^{(k)}$ to generate a skill dataset tailored to **Bob**'s current status. This iterative framework ensures that **Bob** expands the necessary skills at each iteration without any human intervention, supporting the goal of open-endedness.

## 3 EXPERIMENTS

In this section, we present experimental results addressing four research questions:

**RQ1:** How effective is **EXIF** in enabling **Bob** to solve more tasks in the environment by expanding its skill set without human guidance?

**RQ2:** How important is the exploration-first approach in generating valid tasks for **Bob**?

**RQ3:** How do feedback and iterative refinement influence the skill discovery process?

**RQ4:** Can **EXIF** effectively be a self-evolving agent system?

## 3.1 EXPERIMENT SETTINGS

We describe our experimental settings, including environments, models, and baselines. Details are provided in Appendix B (environments), Appendix D (prompts), and Appendix E (implementation).

**Environment**    To answer our research questions, we experiment on two challenging benchmarks with distinct task properties: Webshop (Yao et al., 2022) and Crafter (Hafner, 2022).

- **Webshop**: Webshop is a text-based simulated e-commerce web environment where agents must navigate web pages to purchase a product specified by a natural language instruction. The observation space consists of the textual content of the web pages, and the action space involves searching

queries and clicking UI elements. Key skills include grounding instructions, selecting appropriate search keywords, identifying the correct products, and clicking on the right attributes. This benchmark allows us to evaluate whether using **EXIF** improves Bob's generalization capability when faced with novel products and constraints.

- **Crafter**: Crafter is a Minecraft-like game environment simulating 2D open world. The main objective of the agent in this environment is to survive, explore, gather resources, craft items, and defend against threatening mobs. To interface with LLM agents, we convert image-based observations into a text format by describing the agents' status, inventory, surroundings, and directly facing entities (Paglieri et al., 2025). Key skills in Crafter include exploration, health management, mineral collection, and tool crafting. Within this complex, open-ended benchmark, our aim is to demonstrate that **EXIF**'s goal-less exploration can uncover fundamental skills, like drinking water and collecting resources. Furthermore, we want to show how its iterative feedback loop is crucial for discovering more complex, compositional skills, such as crafting advanced weapons, ultimately enabling the achievement of long-horizon goals.

**Models**    In both experiments, we use `GPT-4o-2024-08-06` (Hurst et al., 2024) as the base LLM for **Alice**. For **Bob**, we employ two different base LLMs: `Qwen2.5-7B` (Yang et al., 2024) and `Llama3.1-8B` (Grattafiori et al., 2024). We also conduct an experiment using the same LLM for both **Alice** and **Bob** (e.g., `Qwen2.5-7B`) to test **EXIF** as a self-evolving system (Section 3.4).

**Baselines**    We compare **EXIF** with several baselines: the proprietary model `gpt-4o` and the base **Bob** models before training. We also evaluate task proposal-first methods (*PF*), where **Alice** proposes tasks without exploration, and rollouts are generated conditioned on these tasks to form the skill dataset. Lastly, we include an explore-first method without a feedback method, denoted as *EF*.

**Exploration Details**    In Webshop, we assign a unique persona for each episode using PersonaHub[1] to encourage diversity. In each round, Alice explores for 250 episodes, ending when a purchase is made or the maximum horizon is reached. In Crafter, Alice is only instructed to survive as long as possible. Each of the 50 episodes ends when the maximum horizon is reached or health points are depleted, following the benchmark's predefined termination criteria.

**Training Details**    As described in Section 2, Alice generates skill dataset to train Bob. In Webshop, we additionally apply post-hoc reasoning (Murty et al., 2024a) to label rationales based on instructions and trajectories. In Crafter, to construct a high-quality skill dataset, we preprocess long-horizon explorative trajectories into segments to generate instructions. While segmenting, we apply a rule-based classifier to monitor changes in the agent's status, inventory, and surrounding entities, but ensure that no additional information is provided beyond the agent's observability. We, then, filter out random and uninformative behavior by retaining only the last four steps of each segment.

**Feedback**    In Webshop, we use **Alice** to provide feedback on **Bob** 's validation performance. Specifically, we use task IDs 501-550 from the validation set. We randomly sample two successful and four failed trajectories, including instructions, based on a reward threshold of 0.5. **Alice** is then prompted to identify model shortcomings and suggest two exploration guidelines as feedback. In Crafter, we request **Bob** to survive in the environment as long as possible without specific goals, mirroring the standard test setup (Paglieri et al., 2025), due to the absence of validation tasks. Then, we prompt **Alice** to generate feedback on **Bob**'s 20 rollout trials in the environment.

**Evaluation**    In Webshop, we utilize the first 500 test tasks to measure the performance of **Bob**. Specifically, we use the environment's predefined reward and the task success rate (*SR*) to measure the performance. In Crafter, we adopt two metrics to thoroughly examine (1) the improvement of skill set and (2) the capability of agents in using the learned skills in a long-horizon interaction with the environment. First, we count the number of learned skills (*NS*) out of 22 pre-defined tasks in the benchmark. When measuring this, we provide an explicit instruction specifying each task and the necessary prerequisites (e.g., the stone pickaxe when mining iron) to the agent, and count the completed skills with at least a 0.5 success rate over 10 trials. Second, we measure the average progress (*AP*) of achievements accomplishments (out of the pre-defined 22 tasks) in a single rollout starting without any prerequisite item, following evaluation of prior work (Paglieri et al., 2025), across 20 trials.

---

[1] https://huggingface.co/datasets/proj-persona/PersonaHub

Table 1: Performance comparison of agents using different base LLMs (`GPT-4o`, `Llama3.1-8B`, `Qwen2.5-7B`) and methods across the Webshop and Crafter environments. *Reward* is the predefined reward in Webshop; *SR* denotes Success Rate in Webshop; *NS* is the number of learned skills in Crafter; and *AP* indicates the average progress rate in Crafter. For Reward, SR, NS, and AP, We report values in the format mean$_{\pm\text{standard error}}$ (improvement over the base model) across multiple evaluations. *# Iter.* refers to the number of iterations conducted in the training process; *# Traj.* indicates the number of trajectories used to train the model throughout the entire training process.

| Base LLM | Method | Webshop | | | | Crafter | | | |
|---|---|---|---|---|---|---|---|---|---|
| | | # Iter. | # Traj. | Reward | SR (%) | # Iter. | # Traj. | NS | AP (%) |
| `GPT-4o` | Base | - | - | $16.5_{\pm1.4}$ | $11.4_{\pm1.2}$ | - | - | 15 | $35.5_{\pm2.4}$ |
| `Qwen2.5-7B` | Base | - | - | $23.2_{\pm1.2}$ | $5.0_{\pm0.1}$ | - | - | 9 | $8.6_{\pm1.5}$ |
| | *PF* | 1 | 1000 | $38.6_{\pm2.4}$ (+15.4) | $6.6_{\pm1.0}$ (+1.6) | 3 | 150 | 10 (+1) | $16.6_{\pm3.0}$ (+8.0) |
| | *EF* | 1 | 1000 | $42.1_{\pm0.2}$ (+18.9) | $6.6_{\pm0.1}$ (+1.6) | 1 | 150 | 11 (+2) | $24.2_{\pm3.3}$ (+14.6) |
| | **EXIF** (Ours) | 4 | 1000 | $\mathbf{50.1}_{\pm0.4}$ (+26.9) | $\mathbf{9.0}_{\pm0.1}$ (+4.0) | 3 | 150 | **15** (+6) | $\mathbf{30.2}_{\pm2.3}$ (+18.8) |
| `Llama3.1-8B` | Base | - | - | $2.0_{\pm0.1}$ | $0.0_{\pm0.0}$ | - | - | 7 | $7.8_{\pm1.2}$ |
| | *PF* | 1 | 250 | $27.2_{\pm2.2}$ (+25.2) | $2.0_{\pm0.0}$ (+2.0) | 3 | 150 | 11 (+4) | $20.1_{\pm3.1}$ (+12.4) |
| | *EF* | 1 | 500 | $38.1_{\pm0.9}$ (+36.1) | $3.0_{\pm0.0}$ (+3.0) | 1 | 150 | 12 (+5) | $25.6_{\pm2.1}$ (+17.8) |
| | **EXIF** (Ours) | 4 | 1000 | $\mathbf{52.6}_{\pm1.2}$ (+50.6) | $\mathbf{6.0}_{\pm0.0}$ (+6.0) | 3 | 150 | **14** (+7) | $\mathbf{31.9}_{\pm3.0}$ (+24.1) |

## 3.2 MAIN RESULTS

**Quantitative Analysis**  Table 1 presents a comparison of agents trained on different datasets for the Webshop and Crafter tasks. Notably, in both tasks, **EXIF** significantly outperforms the base model before fine-tuning, indicating that Alice generates meaningful skill dataset for Bob. Furthermore, compared to *PF* and *EF*, **EXIF** achieves superior performance, highlighting the importance of both the exploration-first strategy and the feedback mechanism.

Specifically, in Webshop, `Llama3.1-8B` model with our method achieves a reward value exceeding 50.0, outperforming the base model (2.0) and even surpassing the proprietary model `GPT-4o` (16.5). The poor performance of `GPT-4o` reflects its unfamiliarity with the Webshop environment and limited ability to solve tasks within a finite horizon. Consequently, *PF* methods perform poorly: Alice not only struggles to complete the proposed tasks but also to generate valid ones, highlighting the need for an exploration-first approach.

Moreover, incorporating a feedback mechanism into *EF*—which is **EXIF**—boosts performance by nearly 50%, underscoring the importance of feedback in guiding the synthesis of training trajectories tailored to the agent. Specifically, as shown in Figure 2, the performance of *EF* plateaus after iteration 1 or 2, whereas **EXIF** exhibits consistent gains due to the feedback mechanism, indicating that naive scaling of data alone does not improve performance. A similar trend is observed for `Qwen2.5-7B`, though in this case, feedback-guided exploration also leads to an increase in success rate.

In Crafter, agents using both `Llama3.1-8B` and `Qwen2.5-7B` achieve performance close to that of `GPT-4o`. Specifically, in the evaluation measuring the number of learned skills, the trained `Qwen` agent matches the base `GPT-4o`, achieving 15 skills out of 22 test tasks. Similarly, the `Llama` agent achieves 14 skills—twice as many as its untrained counterpart. When we evaluate agents by making them survive in the environment for as long as possible without any prerequisite inventory, the `Llama` and `Qwen` agents achieve AP values of 31.9% and 30.4%, respectively. This indicates that the skills discovered by **EXIF** are highly beneficial in long-horizon, open-ended evaluation settings. Compared to the base agents, which average below 9% AP, agents trained with **EXIF** learn to manage health by using resources like food and water, and gradually upgrade their inventory by collecting materials and crafting tools. In contrast, both *PF* and *EF* show limited performance, with AP below 30%, highlighting the advantage of feedback-guided exploration in expanding agent capabilities. Additionally, as shown in Figure 2, the feedback mechanism in **EXIF** enables the agent to learn a greater number of skills (NS) and achieve larger gains in AP over training iterations compared to *EF*, similar to the trend observed in Webshop, highlighting the effectiveness of feedback.

**Qualitative Analysis**  Figure 3 shows qualitative examples demonstrating how, given the same instruction, the trained model differs in its action sequences compared to the base model. In Webshop, we observe that the base model fails to click on attributes such as "size, 21 in x 35 in," whereas after applying **EXIF**, the model successfully follows the instruction by learning how to correctly click

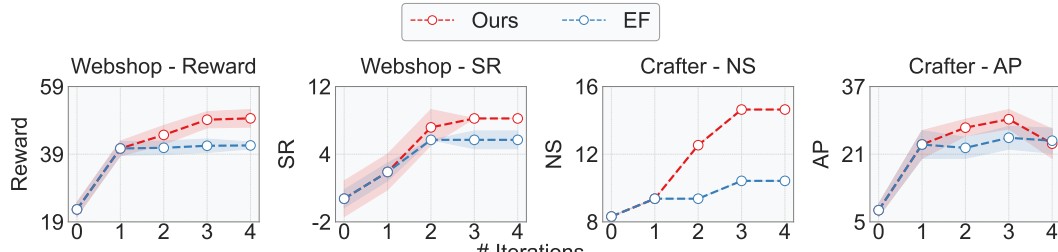

Figure 2: Performance comparison of **EXIF** with feedback at each iteration versus *EF*, which scales data by generating more samples per iteration without feedback, on Webshop and Crafter using `Qwen2.5-7B`. Increasing the amount of data alone does not improve performance without feedback.

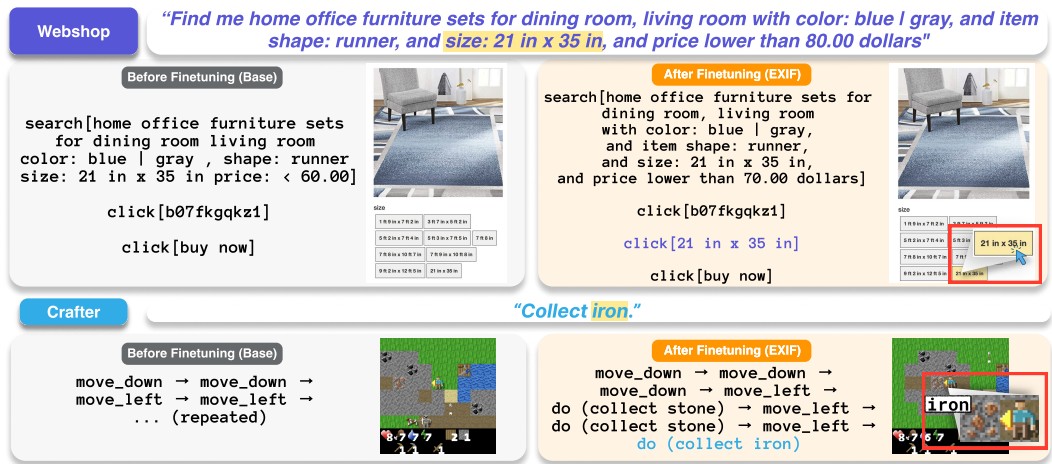

Figure 3: Qualitative examples of action sequences generated by the `Llama3.1-8B` model before and after fine-tuning with **EXIF**. **EXIF** encourages more precise instruction following in the web environment and reduces random behavior or enables new skills in the game environment.

attributes or conditions mentioned in the prompt. In Crafter, the base model exhibits excessive random behavior for the given instruction of "Collect iron". Due to such repetitive behavior, the agents fail to reach the target iron tile as obstructed by the stone tile. On the other hand, the model trained with **EXIF** learns that the skill of collecting stones is necessary to move forward and ultimately reaches the target iron, successfully completing the task. More sample analysis is provided in Appendix H.

## 3.3 TRAJECTORY AND FEEDBACK ANALYSIS

**Proposal-first vs Exploration-first**  A lot of tasks from the proposal-first approach are invalid, as the model proposes goals without precise knowledge of the environment, often leading to infeasible tasks or mismatched trajectories. In contrast, the exploration-first approach yields mostly valid tasks by generating trajectories first and then deriving instructions from the trajectory and final observation, ensuring better alignment. For example, tasks like "Smelt raw beef into cooked beef using coal in the furnace" or "Place a torch in a dark cave area," though seemingly plausible, are indeed invalid in Crafter due to the absence of entities. Figure 4a shows the ratio of valid skill datasets generated by the two approaches: *PF* and *EF*. Specifically, we consider skill data valid if the instruction is feasible in the environment and its trajectory aligns with the corresponding instruction (see Appendix E). We observe that exploration-first methods yield 85% and 70% in Webshop and Crafter, respectively, while proposal-first methods result in less than 30% valid skill dataset, demonstrating the importance of the exploration-first approach for collecting trajectories.

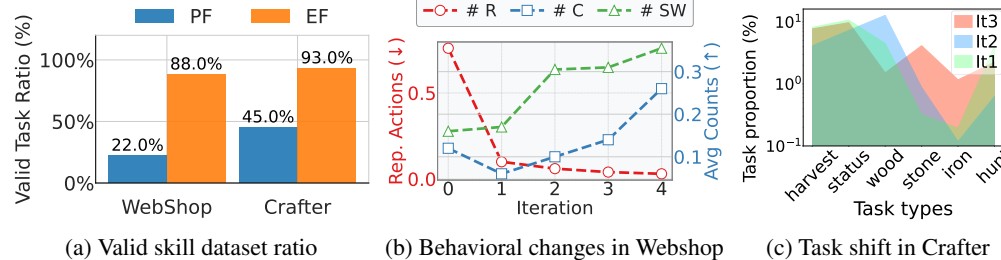

(a) Valid skill dataset ratio  (b) Behavioral changes in Webshop  (c) Task shift in Crafter

Figure 4: **(a)** The ratio of valid skill dataset among those generated using *PF* and *EF* approaches in Webshop and Crafter. **(b)** The average number of repeated actions (# R), average number of clicking attributes (# C), and average number of search keywords (# SW) by **Bob**, normalized by 20 for display, per iteration. **(c)** The skill distribution discovered by **Alice** in each iteration in Crafter.

Table 2: Examples of feedback at each iteration. Critical parts that lead to changes in exploration are highlighted in **bold**.

| Task | Iter. | Feedback |
|------|-------|----------|
| **Webshop** | 1 | 1. The current low reward is due to broad search queries. Use more ... **detailed search keywords** ... during your exploration. 2. The current low reward ... **Avoid clicking the same item** multiple times ... during your exploration. |
| | 3 | 1. The model's initial search query ... **generate a detailed query** that specifies ... like small/medium. 2. The model underutilizes attribute selection. Actively **click on diverse attributes**, ... select specific size options. |
| **Crafter** | 1 | Focus on **practicing stone tool crafting and resource collection** to improve progress on currently underexplored early survival tasks |
| | 3 | Focus on **resource preparation for iron tool crafting**, prioritizing materials that support smelting and tool upgrades; avoid crafting additional wooden tools as they are redundant at this stage |

**Feedback Analysis**  Table 2 presents feedback examples during **EXIF**. In Webshop, early iterations show **Bob** repeating actions and using short queries, while later iterations include feedback prompting attribute interactions (e.g., size, color). As a result, **Alice** adjusts its exploration, and **Bob** exhibits reduced action repetition, increased attribute selection, and more detailed search queries, as shown in Figure 4b. In Crafter, feedback guides **Alice** toward increasingly advanced skills in each round. As shown in Figure 4c, the skill distribution shifts toward tasks targeting different objectives over iterations. Early feedback focuses on basic skills like collecting wood, while later rounds emphasize crafting stone tools, enabling **Bob** to complete more complex tasks (please refer to Appendix G for the definition of each task type).

## 3.4 POTENTIAL OF SELF-EVOLVING SYSTEM

A key strength of **EXIF** is its ability to function as a *self-evolving system*, without requiring larger models for **Alice**. To demonstrate a self-evolving scenario, we set up experiments in which the **Alice** model was replaced from `GPT-4o` to `Qwen2.5-7B`, using the same model for **Bob**.

Surprisingly, as shown in Figure 5, this also leads to a significant performance improvement on both benchmarks compared to the base models, nearly matching the performance of a larger **Alice** model in Webshop and achieving a higher success rate. It also shows comparable performance in Crafter. This suggests that **EXIF** can effectively expand the skill set within the environment without relying on a proprietary model, highlighting the potential of **EXIF** for building a self-evolving system—where two identical agents, without any human intervention, collaboratively generate data and learn to perform well, resembling a form of self-play (OpenAI et al., 2021).

That said, the current open-source 7B model has notable limitations. In Webshop, while performance improves significantly in the first iteration, the feedback mechanism quickly loses impact and performance saturates, as the model struggles to generate useful feedback. In Crafter, which demands

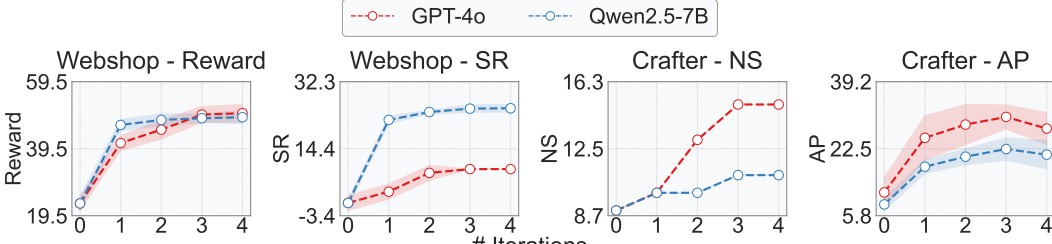

Figure 5: Performance of Bob using the `Qwen2.5-7B` model when Alice is `Gpt-4o` (red) or the `Qwen2.5-7B` (blue) model, investigating the potential of a self-evolving system (blue).

more advanced skills, using `Qwen2.5-7B` as **Alice** further reveals these shortcomings. This implies that models with sufficient capability to diagnose **Bob**'s weaknesses would allow **EXIF** to function as a truly self-evolving system. More analysis and `Llama3.1-8B` results are in Appendix F.

## 4 RELATED WORK

**Curriculum Generation for Autonomous Agent** A line of research has explored methods for automatically generating goal states (Florensa et al., 2018; Pong et al., 2019) or designing training environments (Justesen et al., 2018; Wang et al., 2019; Dennis et al., 2020), enabling agents to continuously learn novel behaviors in open-ended environments. Several works have also investigated self-play approaches (Liu et al., 2019; Team et al., 2021), where agents improve their capabilities by learning to achieve challenging goals generated by their opponents. More recently, LLMs have been used to define curricula (Yang et al., 2023; Du et al., 2023; Nam et al., 2023), and some studies leverage this to create training curricula based on the notion of interestingness (Zhang et al., 2023; Faldor et al., 2024). Additionally, there are works that use LLMs to generate tasks based on the agent behavior or introduces context-aware task proposals (Khan et al., 2024; Zhou et al., 2024b). In this work, we study ensuring the feasibility of the generated plans by letting the LLM explore the environment and, then, relabeling the collected exploration trajectory retroactively.

**Dataset Synthesis for LLM agent** To learn diverse skills, synthesizing datasets with a variety of instructions is crucial. Early approaches to collecting datasets for training LLM agents relied on human annotation (Deng et al., 2023; Lù et al., 2024). Due to the prohibitive cost of manual labeling, AutoWebGLM (Lai et al., 2024) leveraged LLMs to synthesize instructions, while OpenWebVoyager (He et al., 2024b) used LLMs to collect additional trajectories that follow the instructions. Kuba et al. (2025) created a benchmark to test whether expert demonstrations benefit long-horizon tasks. Recently, several works have explored self-improvement through data proposal and iterative interaction, often framed as a two-agent setup, for diverse tasks (Liang et al., 2024; Zhao et al., 2025; Kuba et al., 2025; Chen et al., 2025). In agentic domains, to improve the quality of generated instructions, BAGEL (Murty et al., 2024b) refines synthesized instructions by evaluating agent performance with those instructions. Furthermore, NNetnav (Murty et al., 2024a) and Explorer (Pahuja et al., 2025) propose exploration-based dataset generation, ensuring the feasibility of collected trajectories. Building on these works, our approach extends exploration-based dataset synthesis by introducing iterative interactions between teacher and student agents, enabling more scalable trajectory generation.

## 5 CONCLUSION

We propose **EXIF**, a novel self-improving systems for LLM agents that combines an exploration-first mechanism with iterative training using feedback. Our approach collects trajectories via exploration-guided task generation, uses the explorative agent **Alice** to generate a skill dataset, trains the target agent **Bob** on this dataset, and iteratively refines the exploration strategy based on feedback about **Bob**'s behavior to expand its skill set. Through extensive experiments, we show that the LLM agent's performance improves over multiple iterations, acquiring diverse skills without any human demonstrations—even in a self-play setting. We believe our method represents a meaningful step toward achieving self-evolving systems for open-endedness, enabling agents to autonomously acquire diverse, environment-grounded skills through iterative exploration and feedback.

## REPRODUCIBILITY STATEMENT

For reproducibility, we add details of our settings and implementations in Section 3.1. We add prompts in Appendix D, implementation details are detailed in Appendix E, and we also provide details on environments in Appendix B. We attached the code in our supplementary materials.

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

*Toward Self-Evolving Systems of LLM Agents through Exploration and Iterative Feedback*

# Supplementary Material

## A   LIMITATION & BROADER IMPACT

**Limitations**   While the proposed `EXIF` framework represents a significant step toward self-evolving system, it has some limitations. First, the feedback mechanism, a core component of `EXIF`, relies on natural language, which requires accurate identification of weaknesses. Although it performs well on the benchmarks we evaluated, it may struggle in more complex environments, and the results might vary depending on the capabilities of the base models. Second, we have not explored a version incorporating more predefined skill sets, as in Khan et al. (2024). We plan to extend our work to these more diverse feedback settings and additional environments.

**Broader Impact**   The development of `EXIF` and similar autonomous skill discovery methods holds considerable broader impact for the advancement of artificial intelligence. By enabling agents to autonomously explore, learn, and continuously expand their capabilities without direct human intervention, this research paves the way for a new generation of more independent and adaptive AI systems. Such systems could revolutionize various domains beyond game playing and GUI manipulation, potentially leading to breakthroughs in scientific discovery, personalized education, and complex problem-solving in dynamic real-world scenarios. The ability of agents like `Bob` to generalize to unseen tasks based on self-generated, environment-grounded experiences could significantly reduce the reliance on costly human-annotated datasets, accelerating the deployment of capable AI in a wider array of applications and fostering the creation of truly intelligent systems that can adapt and grow with minimal human guidance.

## B   ENVIRONMENT DETAILS

### B.1   WEBSHOP

We explain the details of the Webshop environment, covering the observation space, action space, the instructions used, and how the benchmark score is calculated.

**Observations**   The observation is a text-based web page, which can be a search page, a product list page, or an item description page. An example of a product list page is detailed below:

---

**Example of Webshop Observation**

[button] Back to Search [button_]

Page 1 (Total results: 20)

[button] Next > [button_]
[button] B09J5HJ8DL [button_]
TASYL USB Adapter for iPhone iPad Lightning Camera Adapter USB 3.0 OTG Cable Supports Camera, USB Flash Drive, Keyboard, Mouse, Camera, Wireless dongles, Bluetooth Dongles $13.8
[button] B07YCGBPRD [button_]
OTAO Privacy Screen Protector for iPhone 11 Pro Max/iPhone Xs Max 6.5 Inch True 28°Anti Spy Tempered Glass Full-Coverage (2-pack) $9.98

. . .

[button] B07DGXZJ1K [button_]

---

> Afeax Compatible Volume Button Silent Power Switch Flex Cable Replacement for iPhone 8 Plus (5.5 inch) $8.9

**Action Space**    Actions consist of two distinct types: search and click. The search action allows the agent to search for items in the web environment and is only available on the initial page with the search button. Search queries can include any keywords related to various products, such as phones, tablets, shoes, clothes, and more.

All actions beyond the initial page are click actions. There are three types of click actions:

- **Clicking HTML elements,** mostly item IDs, to navigate to specific product pages.
- **Clicking navigation options,** where the agent can choose to go back to the previous page, proceed to the next page, return to the search page, purchase a product, etc.
- **Selecting product attributes,** such as color or size, to finalize the product details before purchase.

**Benchmark Evaluation**    For Webshop, there are predefined tasks identified by task IDs. Following the original setting (Yao et al., 2022), we use task IDs 0–499 as evaluation tasks. The instruction in each evaluation task typically takes the form of a search request with specific constraints, such as: "Find me double sided, machine washable decorative pillows with printing technology with size: 28" x 28", and price lower than 30.00 dollars." Each task has a predefined reward based on how similar the selected product is to the ground-truth answer. A success is counted when the reward is 1.0, indicating a perfect match.

### B.2  CRAFTER

We explain the details of the Crafter environment, including the observation space, action space, instruction set, and evaluation setting.

**Observations**    Within our experimental setup, we convert raw image observations into structured textual representations to interface with the LLM agent. Each textual observation encodes the agent's current status, inventory, immediate surroundings, and the entity directly in its line of sight. A specific example is provided below.

---

**Example of Crafter Observation**

### Current Observation
Your status:
- health: 5/9
- food: 8/9
- drink: 9/9
- energy: 8/9

Your inventory:
- wood_pickaxe: 1
- stone: 9
- stone_pickaxe: 1
- coal: 3
- iron: 1
- wood_sword: 1
- stone_sword: 1

You see:
- water 2 steps to your west
- grass 1 steps to your south
- stone 3 steps to your east
- path 1 steps to your east

---

> - sand 1 steps to your west
> - coal 5 steps to your north-east
>
> You are facing path at your front (east direction)

**Action Space** The environment exposes an 17-action discrete control space that can be grouped into five functional categories. **Navigation** actions allow single-tile movement in the four cardinal directions, supporting spatial exploration. **Interaction** enables direct engagement with the forward tile, including resource collection, and combat. **Placement** actions let the agent deploy terrain-modifying objects—stone blocks, crafting tables, furnaces, and plants—that serve as prerequisites for later tasks. **Crafting** actions synthesize tools and weapons when contextual requirements (nearby table or furnace) and inventory resources are satisfied. Finally, **rest/idle** actions restore internal energy or deliberately suspend activity, preserving the agent's state.

- **Navigation**: `move_left, move_right, move_up, move_down`

- **Interaction**: `do`

- **Placement**: `place_stone, place_table, place_furnace, place_plant`

- **Crafting**: `make_wood_pickaxe, make_wood_sword, make_stone_pickaxe, make_stone_sword, make_iron_pickaxe, make_iron_sword`

- **Rest / Idle**: `sleep, noop`

**Evaluation** We evaluate our method in the Crafter environment using two complementary metrics that capture (1) the diversity and number of skills acquired, and (2) the agent's ability to use these skills in long-horizon interactions without task instruction.

- **Number of learned skills (NS)** : To assess the breadth of the acquired skill set, we compute the number of learned skills, denoted as *NS*, out of the 22 pre-defined tasks in the Crafter benchmark. For each task, we provide the agent with an explicit natural language instruction that clearly specifies the goal and any necessary prerequisites. The agent is evaluated over 10 independent trials per task. A task is considered successfully learned if the agent achieves a success rate of at least 0.5 across these trials. This metric reflects the agent's ability to master individual skills when prompted with clear instructions. All trials are conducted using environment seeds `42+i`, where $i = 0, 1, \ldots, 9$.

- **Average progress (AP)** : To evaluate the agent's ability to autonomously achieve goals in an open-ended setting, we compute the average progress, denoted as *AP*. This metric measures the average proportion (ranging from 0 to 1) of distinct achievements accomplished in a single episode, out of the same set of 22 tasks. Following prior work, the agent is initialized without any prerequisite items (i.e., no tools and resources) and runs for one full rollout. The AP score is averaged over 20 such episodes. Unlike NS, which evaluates isolated skill execution under guided instructions, AP captures how well the agent can compose and utilize previously learned skills to make progress toward multiple goals in a long-horizon, unguided setting. All episodes are conducted using environment seeds `42+i`, where $i = 0, 1, \ldots, 19$.

## C ALGORITHM

Algorithm 1 presents the detailed procedure of **EXIF**, with further explanation provided in Section 2.

---

**Algorithm 1: EXIF**: Self-Evolving Systems via Exploration and Iterative Feedback

---

1: **Initialize:**
2:     LLM agent **Alice** (policy $\pi_\phi$ parameterized by $\phi$)
3:     Target LLM agent **Bob** (policy $\pi_\theta$ parameterized by $\theta$)
4:     Total number of iterations $K_{iter}$
5:     Feedback $F^{(-1)} \leftarrow$ null           ▷ *No feedback for the first iteration ($k = 0$)*
6:     $\mathcal{D}_{skill}^{(k)} \leftarrow \emptyset$                            ▷ *Initialize Skill Dataset*
7:
8: **for** $k = 0$ **to** $K_{iter} - 1$ **do**
9:     // — Iteration $k$ —
10:     **// Step 1: Exploration & Skill Dataset Generation**
11:     **if** $k = 0$ **then**
12:         **Alice** explores environment: $a_t \sim \pi_\phi(\cdot \mid h_t, o_t)$     ▷ *Initial exploration phase*
13:         Collect $M$ initial exploratory trajectories $\mathcal{D}_{exp}^{(k)} = \{\tau_{exp}^{(j)}\}_{j=1}^M$
14:     **else**
15:         **Alice** explores environment using feedback $F^{(k-1)}$: $a_t \sim \pi_\phi(\cdot \mid h_t, o_t, F^{(k-1)})$    ▷ *Exploration with feedback*
16:         Collect $M$ targeted exploratory trajectories $\mathcal{D}_{exp}^{(k)} = \{\tau_{exp}^{(j)}\}_{j=1}^M$
17:     **end if**
18:
19:     **// Instruction generation from collected trajectories**
20:     **for** each trajectory $\tau_{exp}^{(j)} \in \mathcal{D}_{exp}^{(k)}$ **do**
21:         **Alice** analyzes $\tau_{exp}^{(j)}$ and generates a natural language instruction $I^{(j)}$
22:         $\mathcal{D}_{skill}^{(k)} \leftarrow \mathcal{D}_{skill}^{(k)} \cup \{(I^{(j)}, \tau_{exp}^{(j)})\}$
23:     **end for**
24:
25:     **// Step 2: Training Target Agent Bob**
26:     Fine-tune **Bob**'s policy parameters $\theta$ to $\theta^{(k)}$ using $\mathcal{D}_{skill}^{(k)}$, yielding policy $\pi_{\theta^{(k)}}$
27:         Minimize SFT loss: $\mathcal{L}_{SFT}(\theta^{(k)}; \mathcal{D}_{skill}^{(k)}) = -\sum_{j=1}^M \sum_{t=1}^{T_j} \log \pi_{\theta^{(k)}}(a_t^{(j)} \mid h_t^{(j)}, o_t^{(j)}, I^{(j)})$
28:
29:     **// Step 3: Evaluation & Feedback Generation**
30:     Evaluate **Bob**'s current policy $\pi_{\theta^{(k)}}$ in the target environment. Let $E_k$ be the evaluation data ▷ *Collect $(o_t, a_t)$, etc.*
31:     **Alice** analyzes **Bob**'s performance $E_k$ to generate natural language feedback $F^{(k)}$ ▷ $F^{(k)}$ *for next iter. (if $k < K_{iter} - 1$)*
32:
33: **end for**

---

## D EXPLORATION PROMPTS

We provide the detailed prompts that are used for the experiment. We used several different types of prompts for each benchmark we used: Webshop and Crafter. The prompts comprise an exploration prompt, an instruction generation prompt, an evaluation prompt, and a feedback generation prompt. In Webshop, we additionally use a post-hoc reasoning prompt.

### D.1 WEBSHOP

#### D.1.1 EXPLORATION PROMPT

---

**Exploration Prompt**

You are a web-shop-agent that can interact with the webpage by taking actions. You need to buy something that you want at the end. Also, you should adopt the identity of following persona :
{task_state.persona}
You should take actions that are consistent with the persona you have adopted.

In the web environment, your actions are strictly limited to two types:

1. search[keywords]: Use this action only when a "[button] Search [button_]" is present in the current web page content. You must replace "keywords" with any valid search query you want to search.

2. click[HTML Element]: Use this action to click on an HTML Element in the page content. "HTML Element" can be any clickable element in the page represented inside "[button]" and "[button_]", such as an item id, action button, or attributes and options like color or size. Note that the 'HTML Element' must be present in the current page content. Also, do not click the attributes inside the "[clicked button]" and "[clicked button_]", "item name", and "button" iteself (e.g. click[button] is not allowed).

Only use search action when a "[button] Search [button_]" is present in the current web page content and otherwise, use click action (click item id, attributes like color and size, or action button).
Feedback from Previous Round :

{feedback_from_alice}
Now here is the new page content. Read carefully the page content. Based on your persona and the current web page content, give a brief thought and provide any valid action that seems very interesting. When outputting the action, please write your action after the prompt 'Action:'.

---

#### D.1.2 INSTRUCTION GENERATION PROMPT

---

**Instruction generation Prompt**

You are a helpful assistant trained to understand web environment and generate shopping instructions. You are given an action sequence and a final product description. Your task is to generate only an user query that will lead to the final product description.

Now here are the given action sequence and final product description.
Action Sequence:
action_sequence

---

Final Product Description:
{final_state}

Considering both search keywords and product detail, please generate an user query. Please put more weight on the search keywords than the product detail. Do not directly include the product name in the query and rather give a high-level description of the product. Note that clicked attributes in action sequence, like size, color, and options should be included in the query. (Buy now is not an attribute)
Attributes without [clicked button] should not be included in the query, as they are not part of the product.
You should also include the price condition in the query (e.g. price lower than XX dollars).
You should not include any other text than the query. Randomly start the query with words "Find me", "Show me", "I am looking for", "I need", "I want", or similar words.

User Query:

### D.1.3   EVALUATION PROMPT

**Evaluation Prompt**

You are an agent with a strict task of completing a web shopping assignment based on the page content and the user's instructions.

In each step, your actions are strictly limited to two types:

1. search[keywords]: Use this action only when a "[button] Search [button_]" is present in the current web page content. You must replace "keywords" with any valid search query you want to search.

2. click[HTML Element]: Use this action to click on an HTML Element in the page content. "HTML Element" can be any clickable element in the page represented inside "[button]" and "[button_]", such as an item id, action button, or attributes and options like color or size. Note that the "HTML Element" *must* be present in the current page content. Also, do not click the "clicked button" or "item name".

Only use search action when a "[button] Search [button_]" is present in the current web page content and otherwise, use click action (click item id, attributes like color and size, or action button). Now, here is the task
Task : {task_name}

To complete the given task, you have taken the following actions:
{action_summary}

Now here is the new page content. Read carefully the page content. Based on the previous actions, the given task, and the current web page content, give a brief thought and provide a valid action. When outputting the action, please write your action after the prompt "Action:".

### D.1.4   FEEDBACK GENERATION PROMPT

**Feedback Generation Prompt**

You are an AI assistant tasked with analyzing web shopping trajectories. To get a high reward, the model needs to complete the task with the given instruction, fulfilling the task requirements of product type, price, attributes like size and color, etc.

Given trajectories of varying rewards, identify strengths in successful trajectories and weaknesses in failed trajectories. Provide concise feedback (2 points maximum) on what skills need improvement to achieve a high reward.

Using your feedback, you will explore the web shopping task on the next round, where your trajectories will be used to train the model. For example, if the model lacks detailed search queries, you need to make an initial query very detailed when the search page is shown because your search will be used as data for fine-tuning the model. Now here are the trajectories of the current model:

{trajectory}

————————————————————

Based on these trajectories, provide concise feedback (2 points maximum) on what kinds of behaviors are desirable and undesirable during exploration. Keep the points very brief.

Most importantly, for each point, write a brief guide on what you need to do during your exploration of the web shopping task on the next round.

Also, you can take up to 10 actions in the environment, so please give feedback on how to have a good and concise action sequence.

*****Note that during your exploration, there are "no instructions, given criteria, or requirements to follow", so you need to provide feedback on which types of actions are beneficial (as there are only two types: search and click, specify which search keywords or clicking on which elements are beneficial).

If you do certain actions with your interest, the models are encouraged to do more of that action.

Thus, do not say something like "do something to meet criteria", "follow the criteria, instructions, or given states", or "match specific attributes". Just say what you think is good or bad.

The example format could be like this:

1. The current low reward is due to B (e.g., limited search). Refrain from B during your exploration.

2. The current low reward is due to not clicking C. Ensure to click diverse C during your exploration.

### D.1.5 POST-HOC REASONING PROMPT

**Post-hoc Reasoning Prompt**

You are an AI assistant tasked with explaining actions taken in a web environment.

Given the instruction you need to follow and the current observation, provide a rationale for why the "last action" was taken to follow the instruction.
You can also refer to the previous actions to provide a rationale.
The rationale should naturally fit with "[your rationale]. Thus, my action is [chosen action]."
You only need to provide "your rationale" part. Be very concise and clear.

Now, here are the given instruction, previous actions, current observation, and the last action.

Instruction: {instruction}

Previous actions before the last action: {previous_actions}

Current observation: {current_observation}

Last action taken based on the current observation: {action}

Why was this last action taken? Provide a rationale:

## D.2 CRAFTER

### D.2.1 EXPLORATION PROMPT

**Exploration Prompt**

You are an intelligent agent navigating and surviving in the Crafter game world while performing the given task, learning and adapting through feedback. Below are the only valid actions you can take in the game, along with their descriptions.

### Valid Actions
- move_left: move one tile west
- move_right: move one tile east
- move_up: move one tile north
- move_down: move one tile south
- do: interact with the tile in front (collect material, drink from lake to restore 'drink' level, attack creature, hunt cow to restore 'food' level)
- sleep: sleep to restore 'energy' level
- place_stone: place a stone in front
- place_table: place a wooden crafting table in front, used for making tools and weapons.
- place_furnace: place a stone furnace in front, used for crafting advanced tools and materials.
- place_plant: place a plant in front
- make_wood_pickaxe: craft a wood pickaxe, which requires a nearby table and wood in your inventory.
- make_wood_sword: craft a wood sword, which requires a nearby table and wood in your inventory.
- make_stone_pickaxe: craft a stone pickaxe, which requires a nearby table and both wood and stone in your inventory.
- make_stone_sword: craft a stone sword, which requires a nearby table and both wood and stone in your inventory.
- make_iron_pickaxe: craft an iron pickaxe, which requires both a nearby table and furnace, as well as wood, coal, and iron in your inventory.
- make_iron_sword: craft an iron sword, which requires both a nearby table and furnace, as well as wood, coal, and iron in your inventory.

### Instructions
- Plan progressively based on your inventory: Before choosing your next action, carefully examine your current inventory. Reflect on the resources and tools you've gathered so far to determine the next meaningful step—whether it's crafting a new tool, upgrading existing gear, or preparing for a more advanced objective.
- Identify and avoid meaningless actions: Each turn you are shown the observation and status from the previous step. Always compare them with the current values; if they are identical, your last move was meaningless—adapt your plan so you do not repeat it.

- Stay alive: When any health falls below its average level, prioritize eating, drinking, sleeping, or defending as appropriate.
- Use the right tools: Some blocks (e.g., stone, iron, diamond) cannot be harvested with a bare hand—craft and equip the correct pickaxe before using do.
- Placement rules: You may place a work table, furnace, plant, or stone only when you are facing a tile of grass, path, or sand.

### Feedback from Previous Round
{feedback_from_alice}

We include the **Feedback from Previous Round** part without the first exploration, by replacing {feedback_from_alice} into appropriate text, such as "- Advance in the Crafter world by strategically collecting resources, crafting tools, and overcoming environmental challenges.".

### D.2.2 INSTRUCTION GENERATION PROMPT

**Relabel Prompt**

You are a language model trained to analyze agent behavior in the game Crafter. Your task is to infer the most likely instruction the agent was pursuing, given a sequence of environmental observations and actions.

**Guidelines:**
- Pay special attention to the most recent observation and action, as they reveal the agent's immediate intention.
- The agent can only interact with the tile it is directly facing, so consider only the facing tile when interpreting interaction actions.
- The do action means the agent is trying to interact with the tile it is facing. For example:
- If facing material: collect material
- If facing grass: collect sapling
- If facing water: drink to restore thirst
- If facing hostile creature: defeat the creature
- If facing cow: hunt to restore hunger
- If there's a table or furnace nearby and your action starts with 'make', you're making a tool. Focus on that action.
- Avoid vague or generic explanations. Be precise and grounded in the recent context.

Your output should clearly state the inferred goal the agent was pursuing, based strictly on its behavior and what it was facing. Keep your response very brief - around 10 words maximum.

Here is a sequence of actions and current observation-action pair the agent took in the Crafter game. The turns are listed in chronological order, from oldest to most recent.

### D.2.3 EVALUATION PROMPT

**Evaluation Prompt**

You are an intelligent agent navigating and surviving in the Crafter game world while performing the given task, learning and adapting through feedback.
Below are the only valid actions you can take in the game, along with their descriptions.

### Valid Actions
- move_left: move one tile west
- move_right: move one tile east
- move_up: move one tile north
- move_down: move one tile south

- do: interact with the tile in front (collect material, drink from lake to restore 'drink' level, attack creature, hunt cow to restore 'food' level)
- sleep: sleep to restore 'energy' level
- place_stone: place a stone in front
- place_table: place a wooden crafting table in front, used for making tools and weapons.
- place_furnace: place a stone furnace in front, used for crafting advanced tools and materials.
- place_plant: place a plant in front
- make_wood_pickaxe: craft a wood pickaxe, which requires a nearby table and wood in your inventory.
- make_wood_sword: craft a wood sword, which requires a nearby table and wood in your inventory.
- make_stone_pickaxe: craft a stone pickaxe, which requires a nearby table and both wood and stone in your inventory.
- make_stone_sword: craft a stone sword, which requires a nearby table and both wood and stone in your inventory.
- make_iron_pickaxe: craft an iron pickaxe, which requires both a nearby table and furnace, as well as wood, coal, and iron in your inventory.
- make_iron_sword: craft an iron sword, which requires both a nearby table and furnace, as well as wood, coal, and iron in your inventory.
- noop: do nothing

### Instructions
- Plan progressively based on your inventory: Before choosing your next action, carefully examine your current inventory. Reflect on the resources and tools you've gathered so far to determine the next meaningful step—whether it's crafting a new tool, upgrading existing gear, or preparing for a more advanced objective.
- Identify and avoid meaningless actions: Each turn you are shown the observation and status from the previous step. Always compare them with the current values; if they are identical, your last move was meaningless—adapt your plan so you do not repeat it.
- Stay alive: When any health falls below its average level, prioritize eating, drinking, sleeping, or defending as appropriate.
- Use the right tools: Some blocks (e.g., stone, iron, diamond) cannot be harvested with a bare hand—craft and equip the correct pickaxe before using do.
- Placement rules: You may place a work table, furnace, plant, or stone only when you are facing a tile of grass, path, or sand.

Now, here is the task
Task : {task_name}

For **NS evaluation**, the agent is prompted with a specific task name (e.g., "Make stone pickaxe"), whereas for **AP evaluation**, the task instruction is replaced with a general open-ended prompt: "Advance in the Crafter world by strategically collecting resources, crafting tools, and overcoming environmental challenges."

### D.2.4  FEEDBACK GENERATION PROMPT

**Feedback Generation Prompt**

You are an expert evaluator analyzing agent behavior in a survival crafting game called Crafter. You will be given a **reduced version** of the agent's trajectory, focusing only on segments where the agent's status and inventory have been changed.

Your output **must** be a JSON object with the following two fields:

```
{
  "behavior_analysis": "Describe what the agent has accomplished
  so far. Mention specific achievements (e.g., placing a table)
  and what those imply about the agent's current progression
```

```
    or intent.",
    "next_iteration_advice": "Suggest a specific, actionable next
    step for  the agent that would likely improve
    its capabilities or unlock new achievements. The advice
    should always start with 'Focus on...' and be
    written as a single sentence.
    It should reflect the agent's current progress
    and identify a meaningful, skill-expanding next goal."
    }

    Guidelines:
    - Do not include any explanation or text outside of the JSON block.
    - Do not list step-by-step logs or inventory diffs — summarize behavior abstractly.
    - Consider the agent's current resources and abilities to suggest realistic next goals.
    - Make sure the 'next_iteration_advice' sentence is specific and skill-oriented, not vague.

    Note: This is a **partial trajectory**, so analyze only what is visible.
```

## E  IMPLEMENTATION DETAILS

### E.1  WEBSHOP

**Exploration**   During exploration, we run 250 episodes per round. Each episode has a maximum horizon of 10 steps. We only retain trajectories that end with a "buy now" action within this limit. During exploration, we provide previous actions but omit previous observations, as they may distract **Alice**'s decision-making. Additionally, we exclude search keywords from the previous actions to prevent the trajectory from resembling a proposal-based approach, where **Alice** would try every option to match the search keywords.

**Training**   We train **Bob** for a maximum of 200 steps with a total batch size of 64. We use the AdamW optimizer with a learning rate of $2e-5$ and a weight decay of 0.01. We utilize LoRA adapters with a rank of 64. Training is performed on NVIDIA A6000 GPUs using DeepSpeed Stage 3 configuration. In Webshop, the model is trained from scratch at each iteration, as continuing from the previous checkpoint may hinder performance—especially when increasing the number of rounds—since excessive training might lead to loss of generalizability.

### E.2  CRAFTER

**Exploration**   During exploration, we run 50 episodes per round, each with a maximum horizon of 100 steps. To collect a diverse set of task-relevant trajectories, each episode is initialized with randomized agent status and inventory configurations, constrained to ensure logical consistency (e.g., we exclude states where the agent possesses a stone pickaxe without having crafted or acquired a wood pickaxe). This setup encourages the agent to explore a broad range of achievable skills without relying on unrealistic initial conditions.

**Processing the trajectories**   To construct a high-quality skill dataset, we process the trajectory collected by **Alice**. We first segment the long-horizon trajectory into several segments by using a rule-based classifier. The rule-based classifier monitors the changes in the agent's observation information. Second, when a change is detected at time $t$, we define a skill trajectory as the four most recent observation-action pairs: $(o_{t-3}, a_{t-3}, \ldots, o_t, a_t)$. Alice then labels these segments with corresponding skill instructions. Each iteration yields roughly 1500 observation-action pairs for Bob's training.

**Training**   We train our model using LoRA-based supervised fine-tuning with a rank of 16. The training is conducted for a total batch size of 32 using the AdamW optimizer with a learning rate of $1e-4$. We leverage NVIDIA A6000 GPUs and adopt the DeepSpeed Stage 3 configuration to enable

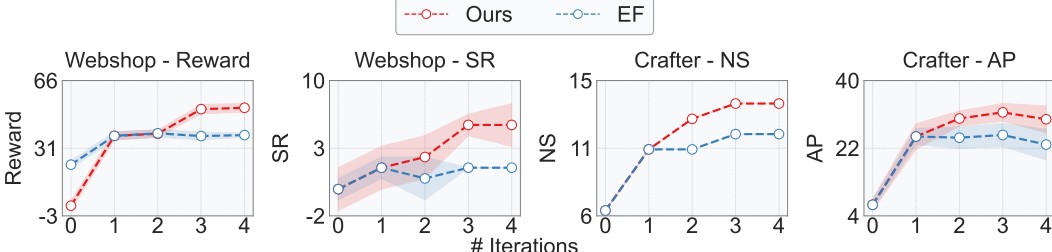

Figure 6: Performance comparison of **EXIF** with feedback at each iteration versus *EF*, which scales data by generating more samples per iteration without feedback, on Webshop and Crafter using `Llama3.1-8B`.

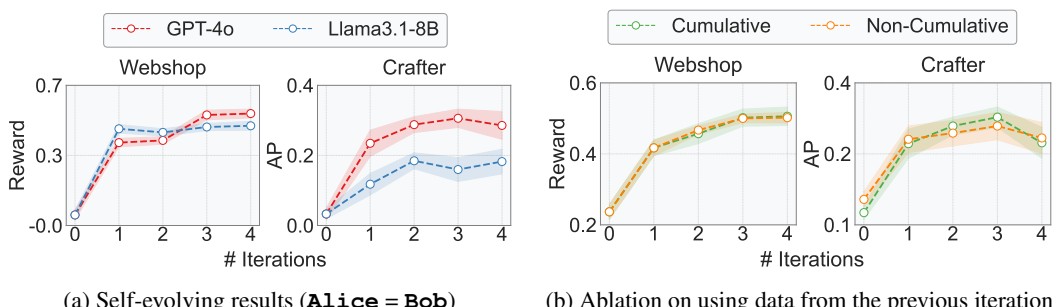

(a) Self-evolving results (**Alice = Bob**)  (b) Ablation on using data from the previous iteration

Figure 7: Performance of Bob using the `Llama3.1-8B` model when Alice is `Gpt-4o` (red) or the `Llama3.1-8B` (blue) model, investigating the potential of a self-evolving system (blue). **(b)** Ablation on whether using data from the previous iteration, where "Cumulative" means using data from previous iterations, and "Non-Cumulative" means not using data from the previous iteration.

efficient large-scale training. We also follow the training scheme in WebShop, where we train the model from scratch at iteration $k$ using the cumulative data up to iteration $k$.

## F  MORE RESULTS

**Results with `Llama3.1-8B`**  Figure 6 compares exploration-first alone and the feedback-based approach, **EXIF**, using the `Llama3.1-8B` model. The feedback mechanism in **EXIF** enables **Bob** to continually improve performance, which cannot be achieved by merely increasing the amount of data in Webshop. A similar trend is observed in Crafter, where feedback allows the agent to acquire more skills and achieve larger gains in AP over training iterations compared to *EF*, underscoring the effectiveness of feedback.

**Self-Evolving Performance of `Llama3.1-8B`**  Figure 7a shows the self-evolving performance of **EXIF** with `Llama3.1-8B`, where both **Alice** and **Bob** use the same model. Exploration-based methods still work, but the feedback mechanism is less effective with this smaller model, especially in Webshop. However, steady improvement is observed in Crafter, highlighting the potential of **EXIF** as a self-evolving system.

**Ablation on Training**  We also conduct an ablation study on data usage to examine whether using the generated dataset from the previous round is beneficial. "Cumulative" indicates using the previous dataset, while "Non-cumulative" means not using it. As shown in Figure 7b, in Webshop, using cumulative data provides limited benefit, since the next iteration produces a higher-quality skill dataset that compensates for what the previous one lacks. In contrast, in Crafter, using cumulative data is more beneficial as a way to prevent forgetting, since the task involves acquiring new skills that are orthogonal to those from earlier rounds, and each generation differs in its skill distribution.

Table 3: Examples of feedback for Webshop and Crafter in the self-evolving scenario with `Qwen2.5-7B`.

| Task | Feedback |
|------|----------|
| **Webshop** | The current low reward is due to not clicking the right products. Click the correct products and attributes during your exploration. |
| **Crafter** | Focus on collecting stone and crafting a stone pickaxe to progress beyond basic wooden tools. |

**Analysis of Self-Evolving Queries**  Table 3 shows the feedback from **Alice** in the self-evolving scenario with `Qwen2.5-7B` models. Unlike feedback from `GPT-4o`, these are much more high-level. Specifically, in Webshop, instructions such as clicking the right products are not suitable for non-goal-conditioned exploration in a web environment. Moreover, compared to the first feedback in Table 2, Crafter feedback contains far fewer skills, resulting in only marginal gains. This suggests that while **EXIF** can function as a self-evolving system, it requires sufficient capability to provide appropriate feedback in order to iteratively improve in an open-ended manner.

## G   DETAILS ON SKILLS

**Webshop**  In WebShop, there are no explicit skills pre-defined in the environment. However, as explained in Section 3.3, certain high-level skills are required to perform well across diverse tasks. These include searching with detailed keywords, navigating the web, backtracking, clicking the correct product, refining search queries, reading descriptions and features, and selecting the appropriate attributes.

As shown in Figure 4b, **EXIF** effectively improves detailed search queries and selects the correct attributes while avoiding unnecessary, duplicate actions. We also expected **Alice** to exhibit advanced navigation behaviors, such as using the next or previous buttons, but found that these behaviors actually harmed performance. In WebShop, navigating further does not necessarily lead to better product discovery. The same holds true for backtracking. We believe that more advanced and meaningful skills will emerge in future, more challenging benchmarks using **EXIF**.

**Crafter**  Unlike WebShop, Crafter allows us to observe explicit skills required for long-term survival through a set of predefined tasks. As shown in Figure 4c, **Alice** discovers more skills with each iteration, which in turn improves **Bob** 's performance over time. We additionally define task types to group the pre-defined skills. The full list of tasks, along with task types and their descriptions, is provided in Table 4.

## H   MORE EXAMPLES

### H.1   WEBSHOP

We provide additional examples of **Bob** 's performance across iterations in WebShop. For better visualization, incorrect actions at each step are highlighted in red, while correct actions are shown in green. The example is presented below:

---

**Comparison of Iteration 1 and Iteration 2 of EXIF in WebShop**

**Instruction:** Find me machine wash men's pants with relaxed fit with color: grey, and size: 40w x 34l, and price lower than 60.00 dollars

**Unsuccessful Trajectory (Iteration 1)** "search[men's pants] → click[b099231v35] → click[buy now]"

**Successful Trajectory (Iteration 2)** "search[machine wash men's pants with relaxed fit, 40w 34l]→ click[b08lkksl8f] → click[grey] → click[40w x 34l] → click[buy now]"

---

Table 4: Task skill categories, the full list of corresponding skills under each category, and descriptions of each skill used in Crafter.

| Task Type | Task Name | Description |
|---|---|---|
| Harvest | collect_sapling | Gather saplings from the grass |
| | place_plant | Place a plant on the ground |
| | eat_plant | Eat a plant to recover health |
| Status | wake_up | Wake up after sleeping |
| | eat_cow | Hunt a cow |
| | collect_drink | Drink water in front of the river |
| Wood | collect_wood | Chop trees to collect wood |
| | place_table | Place a crafting table |
| | make_wood_pickaxe | Craft a wooden pickaxe |
| | make_wood_sword | Craft a wooden sword |
| Stone | collect_stone | Mine stone blocks |
| | make_stone_pickaxe | Craft a stone pickaxe |
| | make_stone_sword | Craft a stone sword |
| | place_stone | Place a stone block in the ground |
| Iron | collect_coal | Mine coal blocks |
| | place_furnace | Place a furnace for crafting advanced tools |
| | collect_iron | Mine iron blocks |
| | make_iron_pickaxe | Craft an iron pickaxe |
| | make_iron_sword | Craft an iron sword |
| | collect_diamond | Mine diamond blocks |
| Hunt | defeat_skeleton | Defeat a skeleton enemy |
| | defeat_zombie | Defeat a zombie enemy |

In this example, at Iteration 1, where **Bob** is trained once using **Alice** 's initial skill dataset, the model generates a less detailed prompt—simply "men's pants"—which results in a poor item choice. In Iteration 2, after training on a skill dataset generated based on feedback, **Bob** improves by conducting a more detailed search and clicking better attributes, successfully following the instruction. However, Iteration 2 **Bob** is still imperfect at attribute selection. By Iteration 3, with feedback emphasizing the need to click more attributes (as shown in Table 2), it finally improves its skill in selecting the correct attributes, as demonstrated in the example below.

---

**Comparison of Iteration 2 and Iteration 3 of EXIF in WebShop**

**Instruction:** Find me slim fit men's henleys with short sleeve with color: 157- green, and size: 3x-large, and price lower than 40.00 dollars

**Unsuccessful Trajectory (Iteration 2)** "search[slim fit men's henleys short sleeve 157 green 3x-large] → click[b09r9ycm6r] → click[buy now]"

**Successful Trajectory (Iteration 3)** "search[slim fit men's henleys with short sleeve in color 157-green, size 3x-large, and price lower than 40.00 dollars]→ click[b09r9ycm6r] → click[157- green] → click[3x-large]' → click[buy now]"

---

### H.2 CRAFTER

We also provide additional examples of **Bob** 's performance across iterations in Crafter. For better visualization, incorrect actions at each step are highlighted in red, while correct actions are shown in green. Navigating actions are shown in black. Below is an example of **Bob** 's improved skill set in Iteration 2, compared to Iteration 0 and Iteration 1.

---

**Comparison of Iteration 0, Iteration 1 and Iteration 2 of EXIF in Crafter**

**Instruction:** make_stone_sword

**Unsuccessful Trajectory (Iteration 0)** "move_right → move_down → make_stone_sword ... *(repeated)*"

**Unsuccessful Trajectory (Iteration 1)** "move_up → move_up → place_table → do → do ... *(repeated)*"

**Successful Trajectory (Iteration 2)** "move_left → move_down → place_table → make_stone_sword""

---

In this example, at Iteration 0, **Bob** fails because it attempts to craft the stone sword without first placing a crafting table. It does not recognize that a table is a necessary prerequisite for crafting. In Iteration 1, **Bob** places the table, but it uses the "do" action repeatedly, which is not sufficient to trigger the specific crafting behavior. This indicates a lack of understanding that crafting requires an explicit "make_stone_sword" action, not a generic interaction. Finally, in Iteration 2, **Bob** correctly identifies both the prerequisite "placing the table" and the appropriate action, which is explicitly calling the "make_stone_sword" action.

Another example is shown below:

---

**Comparison of Iteration 2 and Iteration 3 of EXIF in Crafter**

**Instruction:** make_stone_sword

**Unsuccessful Trajectory (Iteration 2)** "move_right → move_right → do → do ... *(repeated)*"

**Successful Trajectory (Iteration 3)** "move_right → move_right → do → move_left → do → move_up → do""

---

In Iteration 2, **Bob** finds the zombie but repeatedly uses the "do" action without accounting for the zombie's movement. As a result, it fails to make effective contact and cannot defeat the zombie, reflecting a lack of adaptation to dynamic enemy behavior. In contrast, in Iteration 3, **Bob** 's action sequence demonstrates adaptive behavior: **Bob** actively adjusts its position in response to the zombie's movement, tracking the enemy until it successfully defeats it. This indicates an emerging understanding of how to engage moving entities in the environment, highlighting the effectiveness of **EXIF**.

## LLM USAGE

Every part of the research, including the research questions and ideas, originates entirely from the authors. The paper was written manually by the authors, with LLMs used only to check typos and polish minor grammar in some parts.

