# OpenReview forum: "Toward Self-Evolving Systems of LLM Agents through Exploration and Iterative Feedback"
_ICLR.cc/2026/Conference — ICLR 2026 Conference Withdrawn Submission_

### Official Review · Reviewer_UDZZ · 2025-10-26

**Soundness:** 2
**Presentation:** 3
**Contribution:** 2
**Rating:** 2
**Confidence:** 4

**Summary:**

The paper introduces a framework enabling large language model agents to autonomously acquire new skills through environment-grounded exploration and feedback-driven refinement. EXIF employs two agents—Alice (explorer and instructor) and Bob (learner)—where Alice generates skill datasets through exploration, trains Bob, then provides feedback to guide further data generation. This closed-loop process allows continual self-improvement without human intervention. Experiments on Webshop and Crafter show performance gains and expanding skill repertoires.

**Strengths:**

1. The paper is clearly written and logically structured,

2. The proposed EXIF framework is presented with clarity and step-by-step logic, supported by pseudocode, examples, and detailed prompts in the appendix.

**Weaknesses:**

1.  While Webshop and Crafter effectively demonstrate EXIF’s potential, they remain constrained to text-based and game-like domains. To strengthen generalization claims, the authors could extend evaluations to more complex and multimodal environments.

2. There are lack of comparison with state-of-the-art self-improvement or curriculum-generation frameworks such as BAGEL (Murty et al., 2024b), Explorer (Pahuja et al., 2025), or AutoWebGLM (Lai et al., 2024).

3. The novelty of the proposed  method  should be given more detail description. Lots of self-improvement or curriculum-generation frameworks have been proposed.

Fang, Tianqing, et al. "WebEvolver: Enhancing Web Agent Self-Improvement with Coevolving World Model." arXiv.org (2025).
Wang, Zhenhailong, et al. "Mobile-agent-e: Self-evolving mobile assistant for complex tasks." arXiv preprint arXiv:2501.11733 (2025).
Sun, Zeyi, et al. "Seagent: Self-evolving computer use agent with autonomous learning from experience." arXiv preprint arXiv:2508.04700 (2025).

**Questions:**

What's the main novelty of the proposed method?

---

### Official Review · Reviewer_1MqV · 2025-10-29

**Soundness:** 2
**Presentation:** 3
**Contribution:** 2
**Rating:** 4
**Confidence:** 4

**Summary:**

This paper proposes a training method or framework called Exploration and Iterative Feedback (EXIF), designed to train models to perform agent-based tasks. EXIF enables self-learning by introducing two agents — Alice (the explorer) and Bob (the executor). Alice interacts with the environment to generate feasible skill trajectories and corresponding instructions, which are then used to train Bob. After Bob’s training, Alice evaluates Bob’s performance on a validation set and provides feedback to guide its next round of exploration, forming an iterative learning loop. In the experiments presented by the authors, EXIF significantly improves the agents’ task execution abilities in two benchmark environments — Webshop and Crafter. The framework enables the agent to continuously expand its skill set without human intervention. Notably, when Alice is replaced by the same model architecture as Bob, the trained model still performs effectively, demonstrating that this framework facilitates model self-evolution.

**Strengths:**

1. The paper focuses on a cutting-edge issue，which is the training of LLM-based agents and accurately points out the limitations of current data synthesis methods, namely the lack of interaction with real environments and the gap between artificially synthesized data and actual agent applications.
2. The paper’s central idea aligns with practical needs, actively exploring self-evolutionary learning approaches, which represents a valuable attempt toward future model training paradigms.
3. The experimental design of the paper effectively demonstrates the importance of each component within the EXIF framework and adheres to scientific experimental methodology.

**Weaknesses:**

1. **Limited scalability due to validation data dependency:** The EXIF framework's feedback signals are derived entirely from Bob's performance on manually annotated validation sets. While this approach significantly reduces human intervention during training, the dependency on human-curated evaluation data limits scalability, as the system's improvement is bounded by the scope of available validation tasks. As Bob approaches the performance ceiling on the fixed validation set, further progress may require expanding the manually curated evaluation data. Exploring alternative feedback mechanisms that reduce this dependency could enhance the framework's long-term scalability.

2. **Unfair comparison due to different resource costs:** The core exploration, task generation, and feedback mechanisms of the EXIF framework are all accomplished by the Alice model, which makes EXIF essentially dependent on Alice's capabilities—or in other words, it is essentially distilling Alice. Although "EF" demonstrates that methods with feedback are more effective than directly generating trajectories for training, this comparison is actually unfair—because EXIF consumes more tokens, or rather, trades increased resource consumption for improved performance.

3. **Concerns about testing rigor:** Since the data for training Bob comes from the results after Alice explores the environment, this leads to Bob having already seen similar environments during testing. There is one scenario to consider: for example, if a task requires four steps A, B, C, and D, perhaps Alice's exploration has already completed A, B, and C, and this trajectory has been trained into Bob. Then Bob only needs to supplement action D based on A, B, and C, which may be a very simple task for Bob. Therefore, I have concerns about the quality of the test data.

4. **Concerns about Bob's capability degradation:** Bob's evolution comes from SFT, but many studies have already demonstrated that SFT can lead to significant knowledge forgetting [1]. Therefore, I am concerned that the trained Bob will perform poorly in environments outside the training environment. This raises a serious problem: in practical applications, it may be very difficult to have a suitable validation set for Bob's training.

**References:**

[1] Yun Luo, Zhen Yang, Fandong Meng, Yafu Li, Jie Zhou, and Yue Zhang. "An Empirical Study of Catastrophic Forgetting in Large Language Models During Continual Fine-tuning." arXiv preprint arXiv:2308.08747, 2025.

**Questions:**

1. **Regarding "Unfair comparison due to different resource costs":** Could the authors provide a fair comparison under controlled resource budgets? Specifically, I suggest comparing EXIF and EF when both methods use approximately the same computational resources (e.g., total tokens consumed or FLOPs). This would help demonstrate whether the feedback mechanism in EXIF provides benefits beyond simply using more compute.

2. **Regarding "Concerns about testing rigor":** Could the authors provide more detailed examples or statistics about the training and test task distributions? Specifically, it would be helpful to understand: (a) the complexity difference between training and test tasks, (b) concrete examples showing that test tasks require meaningful skill composition rather than simple extensions of training trajectories, and (c) metrics quantifying the overlap between training and test scenarios.

3. **Regarding the concerns about "Limited scalability due to validation data dependency" and "Concerns about Bob's capability degradation":** I am very much looking forward to the authors' perspectives and to discussing with them how they view these issues.

4. **Regarding the "PF" method:** I could not find sufficient details on how training trajectories for Bob are generated in the PF baseline. Specifically, when questions are posed directly based on the environment, how are the corresponding solution trajectories obtained? How do you ensure that the generated questions are solvable within the environment? Clarification on this methodology would be very helpful.

---

### Official Review · Reviewer_yPiN · 2025-10-29

**Soundness:** 2
**Presentation:** 4
**Contribution:** 2
**Rating:** 4
**Confidence:** 4

**Summary:**

This paper aims to build self-evolving systems of LLM agents, to enable open-endedness, and collect suitable training data without human intervention. The authors propose a self-improving framework through EXploration and Iterative Feedback (EXIF). It has two main components: an explore-first strategy that enables the exploration agent (Alice) to navigate the environment and generate feasible, valid tasks, which are then used to train the task-performing agent (Bob); and an iterative feedback mechanism that produces tasks and trajectories beyond Bob's current capabilities to expand its skills. Experiments on Webshop and Crafter demonstrate EXIF's effectiveness to iteratively expand the capabilities of the trained agent.

Overall, this paper is easy to understand, with very clear motivation and presentation. However, the core contributions are less insightful.

**Strengths:**

1. Clear motivation for introducing the EXploration and Iterative Feedback (EXIF) framework.

2. This paper is easy to understand, with a very fluent story flow in writing. The presentations in terms of figures and tables are also quite clear.

3. The EXIF framework is simple yet effective.

**Weaknesses:**

1. The differences between the proposed EXIF method and existing self-evolving methods (e.g., self-play) in LLM agents appear minimal, which limits the novelty and insights of the work.

2. Since the EXIF method involves two main mechanisms in environment exploration and feedback generation, its technical contributions are not yet clear.

3. The experimental design is not convincing enough. Beyond the base models, the evaluation is limited to comparisons with only two variants of the proposed method (i.e., PF and EF), resembling a variant analysis rather than a comprehensive evaluation. A broader comparison with established baseline methods for agent self-evolution is needed to validate the effectiveness of EXIF.

**Questions:**

1. What are the key differences between the proposed EXIF method and many other self-evolving methods (e.g., [1-4]) in LLM agents?
2. As shown in Table 1, prompting GPT-4o achieves the highest task success rate on Webshop and the average progress on Crafter. How about the performance when using GPT-4o as the base LLM? Can GPT-4o further self-improve with the proposed EXIF?

[1] SELF: Self-Evolution with Language Feedback. arXiv'2023.
[2] I-SHEEP: Self-Alignment of LLM from Scratch through an Iterative Self-Enhancement Paradigm. arxiv'2024.
[3] Self-Play Fine-Tuning Converts Weak Language Models to Strong Language Models. ICML'2024.
[4] Interactive Evolution: A Neural-Symbolic Self-Training Framework For Large Language Models. ACL'2025

---

### Official Review · Reviewer_3fPK · 2025-11-01

**Soundness:** 2
**Presentation:** 4
**Contribution:** 2
**Rating:** 2
**Confidence:** 3

**Summary:**

This paper proposes the EXploration and Iterative Feedback (EXIF) framework, which aims to address the feasibility and adaptability issues in generating training data for Large Language Model (LLM) agents. The framework leverages an exploration agent, Alice, to interact with the environment and generate a feasible, environment-grounded skill dataset—avoiding the invalid data associated with the "task-proposal-first" approach—to train a target agent, Bob. Additionally, an iterative feedback loop is introduced: Alice evaluates Bob’s performance and guides targeted exploration in the next round, forming a closed-loop data generation process. Experiments on the Webshop and Crafter benchmarks demonstrate that EXIF continuously enhances Bob’s capabilities without human intervention (e.g., the reward of the Qwen2.5-7B model in Webshop increases from 2.0 to 52.6). Notably, significant performance improvements are still observed when Alice and Bob use the same small model (e.g., Qwen2.5-7B), with a 15% higher success rate in Webshop, verifying EXIF’s potential for building self-evolving systems

**Strengths:**

Unlike the traditional "task-proposal-first" approach (which tends to generate invalid data), the method employs an exploration agent, Alice, that first interacts with the environment to generate feasible trajectories—such as exploring shopping processes based on personas in Webshop and exploring with survival as the goal in Crafter—before reverse-engineering instructions from these trajectories to form a "skill dataset." This "trajectory→task" logic ensures each task has an environment-grounded execution path. Experiments show that the proportion of valid data generated (85% in Webshop and 70% in Crafter) is far higher than that of the task-proposal-first (PF) approach (<30%), fundamentally solving the core problem of invalid training data.

**Weaknesses:**

1. In Crafter, the paper segments long trajectories and retains only the last 4 steps, explaining this as a means to "filter random behaviors." However, it fails to clarify the rationale for choosing "4 steps" as the segment length.
2. In Webshop, the paper chooses to "train Bob from scratch in each iteration instead of fine-tuning based on the previous checkpoint," justifying this with "avoiding reduced generalization due to over-training." Yet, no comparative experimental data is provided (e.g., performance differences between "training from scratch" and "fine-tuning based on checkpoints").
3. In self-evolving experiments, only the dual-model configuration with Qwen2.5-7B is validated; the dual-model performance of other small models (e.g., Llama3.1-8B) is not tested.
4. Although the paper mentions "providing code and appendix details," key experimental information is still missing, affecting reproducibility. For instance, the complete content of "personas" in Webshop, the "post-hoc reasoning prompt template," and the details of the "rule-based classifier for trajectory segmentation" in Crafter are not fully presented in the appendices. Specific parameters for model fine-tuning (e.g., β1 and β2 values of the AdamW optimizer) and detailed settings for environment seeds (e.g., the seed range for Webshop test tasks) are also unclear. Furthermore, the paper lacks variance analysis of experiments (e.g., whether the standard deviation of repeated experiments is stable), and some results (e.g., the reward of Llama3.1-8B exceeding 50.0 in Webshop) lack error range labels, reducing the credibility of the findings.
5. The paper should include more baselines as in [1][2], making it more solid.
6. The paper only study WebShop and Crafter, which are two easy benchmarks. More tasks and environments should be included, as in [1][3].
7. Some other agentic workflows/mechanisms should be included, liks [4][5].

[1] AgentGym: Evolving Large Language Model-based Agents across Diverse Environments
[2] Agent-FLAN: Designing Data and Methods of Effective Agent Tuning for Large Language Models
[3] AgentTuning: Enabling Generalized Agent Abilities for LLMs
[4] Reflexion: Language Agents with Verbal Reinforcement Learning
[5] Agent-R: Training Language Model Agents to Reflect via Iterative Self-Training

**Questions:**

1. In the trajectory processing for the Crafter environment, the paper segments long trajectories and retains only the last 4 steps to filter random behaviors. What is the rationale for selecting "4 steps" as the segment length? Were experiments conducted to compare the impact of different segment lengths (e.g., 3, 5, or 6 steps) on training outcomes—such as differences in the number of skills Bob acquires or average progress? If no such comparison was performed, why is 4 steps considered the optimal segment length?
2. In Webshop experiments, the paper chooses to "train Bob from scratch in each iteration" instead of fine-tuning based on the previous checkpoint, explaining this as "avoiding reduced generalization due to over-training." However, no comparative experimental data between the two training methods is provided. Could you supplement experiments to show the specific differences in Bob’s reward, success rate, and generalization ability to unseen products in Webshop between "training from scratch" and "fine-tuning based on checkpoints" under the same number of iterations and data volume? Additionally, the "train-from-scratch" strategy is also used in Crafter—could you explain whether this strategy poses a risk of "skill forgetting" in Crafter (e.g., whether the previously acquired skill of "collecting wood" degrades in subsequent iterations) and provide verification data?
3. Self-evolving experiments only validate the dual-model configuration where both Alice and Bob are Qwen2.5-7B, without testing other small models (e.g., Llama3.1-8B). When both Alice and Bob are Llama3.1-8B, what are Bob’s performance metrics—such as reward and success rate in Webshop, and the number of acquired skills and average progress in Crafter? Do different small models differ in feedback generation capabilities (e.g., accuracy of weakness identification, specificity of guidance), and how does this difference affect self-evolving performance? Furthermore, after how many iterations does the dual-small-model configuration reach performance saturation, and is this saturation related to the inherent capability limits of the model itself?
4. The paper mentions "providing code and appendix details" to ensure reproducibility, but key information is still missing. Could you fully supplement the following content: the complete list of personas used to guide Alice’s exploration in Webshop (including detailed descriptions and behavioral tendencies of each persona) and the full post-hoc reasoning prompt template; the specific rules of the rule-based classifier for trajectory segmentation in Crafter (e.g., which state/inventory changes trigger segmentation and how "valid behavior segments" are determined); complete parameters for model fine-tuning (e.g., β1 and β2 values of the AdamW optimizer, learning rate decay strategy) and the range of environment seeds used in experiments (e.g., seed numbers for Webshop test tasks and Crafter training/evaluation); and results of repeated experiments for all metrics (including mean values and standard deviations)—with particular attention to adding error range labels for the reward of Llama3.1-8B exceeding 50.0 in Webshop?

---

### Note · Authors · 2026-01-02

I have read and agree with the venue's withdrawal policy on behalf of myself and my co-authors.